

# Concerning the detection of electromagnetic knot structures in space plasmas using the wave telescope technique

Simon Toepfer[1], Karl-Heinz Glassmeier[2,3], and Uwe Motschmann[1]

[1]Institut für Theoretische Physik, Technische Universität Braunschweig, Braunschweig, Germany
[2]Institut für Geophysik und extraterrestrische Physik, Technische Universität Braunschweig, Braunschweig, Germany
[3]Max-Planck-Institut für Sonnensystemforschung, Göttingen, Germany

**Correspondence:** Simon Toepfer (s.toepfer@tu-braunschweig.de)

**Abstract.**

The wave telescope technique is broadly established in the analysis of spacecraft data and serves as a bridge between local measurements and the global picture of spatial structures. The technique is originally based on plane waves and has been extended to spherical waves, phase shifted waves as well as planetary magnetic field representation. The goal of the present study is the extension of the wave telescope technique using electromagnetic knot structures as a basis. As the knots are an exact solution of Maxwell's equations they open the door for a new modeling and interpretation of magnetospheric structures, such as plasmoids.

## 1 Introduction

The classification and mathematical modeling of spatial structures is one of the major missions of theoretical physics. Especially our extraterrestrial space environment provides a diversity of spatial structures with different characteristics. For example, oscillating structures can be classified into plane waves (e.g., MHD-waves), spherical waves generated at the bow shock, surface waves triggered by instabilities at the magnetopause as well as phase shifted waves caused by field line resonances (Plaschke et al., 2008; Narita et al., 2022). On the other hand, global planetary magnetic fields can be interpreted in terms of a multipole series, based on spherical harmonics (Gauss, 1839; Glassmeier and Tsurutani, 2014; Toepfer et al., 2020a, b, 2021). For the characterization of such structures, empirical models, such as magnetospheric models or models based on a set of specific basis functions spanning the solution-space of differential equations, are required.

In general, any spatial structure can be expanded into a set of mathematical basis functions, such as plane waves or spherical harmonics. Plane waves are the simplest spatial structures forming a basis for the representation of spatial fields. The contribution of any plane wave with its characteristic spatial scale to the total field is described by the spectrum of the field. However, in the worst case, infinitely many elements forming the basis have to be incorporated for describing the structure, resulting in an infinite set of expansion coefficients that have to be determined from the measurements. In this case, it is desirable to choose a new representation based on a new set of basis functions that are well-adjusted to the symmetry of the structure with fewer





unknown parameters.

Electromagnetic knots, proposed by Cameron (2018), are a special superposition of infinitely many plane waves, forming such a new basis set for localized, divergence-free structures, namely the electromagnetic ring and the electromagnetic globule. The geometry of these basis elements is depicted in Fig. (1a) and (1b). A variety of electromagnetic field topologies can be constructed by spatially distributing and superposing several rings and globules as illustrated in Fig. (1c). The complexity of

the emerging field geometries prompts the naming *electromagnetic knots* (Cameron, 2018).

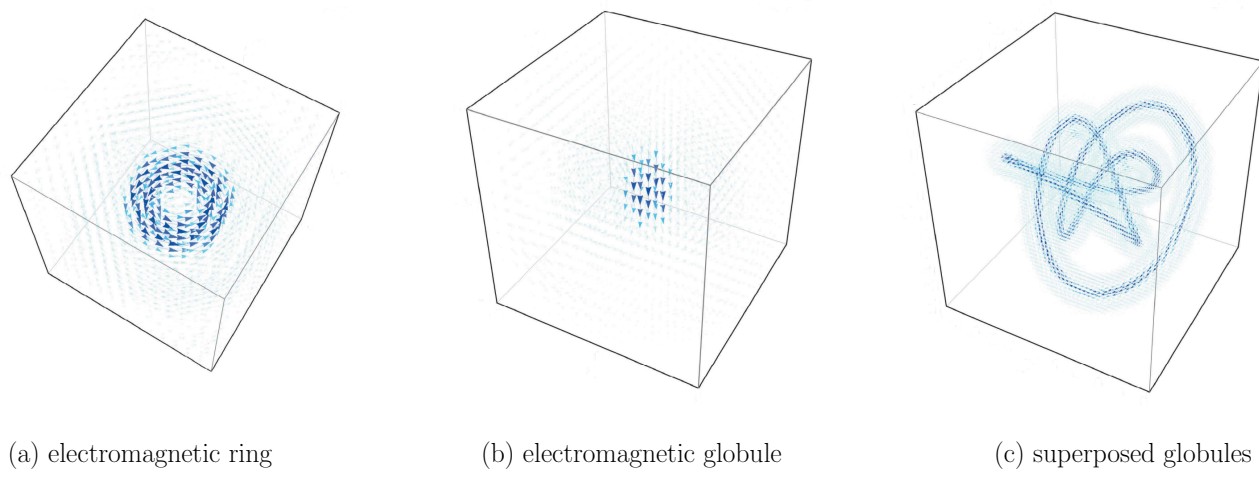

(a) electromagnetic ring            (b) electromagnetic globule            (c) superposed globules

**Figure 1.** Vector representation of the electromagnetic ring (a), the electromagnetic globule (b) and spatially distributed, superposed globules (c) after Cameron (2018).

The electromagnetic ring and the electromagnetic globule are an exact solution of Maxwell's equations and provide a new tool in the context of plasma physical and electrodynamical modeling. Based on the elaboration of Cameron (2018), the mathematical foundations of electromagnetic knots are revisited in the present study. Within this context, the formalism is reformulated in terms of the classical wave telescope technique (Motschmann et al., 1996). Additionally, the applicability of

describing and interpreting spatial structures in planetary magnetospheres via knots is discussed. The wave telescope technique enables the classification of spatial structures in planetary magnetospheres from a limited number of satellite positions and has successfully been applied to several problems in space physics (Glassmeier et al., 2001; Narita et al., 2003, 2009, 2013, 2022). Originally, the method is based on a plane wave representation and was later extended to spherical waves (Contantinescu et al., 2006), phase shifted waves (Plaschke et al., 2008) and planetary magnetic fields (Narita, 2019; Toepfer et al., 2020a, b). The

goal of the present study is the extension of the zoo of spatial structures that can be analyzed from a limited set of measurement positions by considering the electromagnetic knots as a new basis set for the wave telescope. The method is tested against synthetically generated magnetic field data describing a plasmoid as a two-dimensional magnetic ring structure.



## 2 The classical wave telescope

Maxwell's equations represent a set of coupled partial differential equations for the magnetic field $\underline{B}(\underline{x}, t)$ and the electric
$\underline{E}(\underline{x}, t)$, respectively. These equations can be transformed into a set of algebraic equations via the Fourier transform. In the following discussion we will focus on the magnetic field.

The measurement position $\underline{x}$ and the measured field $\underline{B}(\underline{x}, t)$ are known from a set of magnetometer measurements. Due to the high temporal resolution of the magnetometer, the temporal Fourier transform can be applied to the data, delivering the spectral
amplitude $\underline{B}(\underline{x}, \omega)$ (Motschmann et al., 1996). In general, this spectral amplitude is a continous function of $\omega$. However, in the practical application outstanding points of the spectrum, for example sharp maxima, are of major interest. Thus, the data are evaluated at a peak, where $\omega = \omega_0$, with the corresponding amplitude $\underline{B}(\underline{x}, \omega_0)$. So far, the magnetic field can be written as

$$\underline{B}(\underline{x}, \omega_0) = \int \underline{\hat{B}}_0(\underline{k}, \omega_0) e^{i\underline{k} \cdot \underline{x}} \mathrm{d}^3 k, \tag{1}$$

where $\underline{\hat{B}}_0(\underline{k}, \omega_0)$ is the spectral amplitude of the magnetic field with respect to the wave vector $\underline{k}$. As the magnetic field mea-
surements are solely available at a limited number of measurement points, the spatial Fourier transform is not applicable. Thus, the spectral amplitudes $\underline{\hat{B}}_0(\underline{k}, \omega_0)$ and the corresponding wave vectors $\underline{k}$ are to be determined by the data fitting procedure. Although there is a variety of inversion techniques available (Haykin, 2014, e.g.), we will focus on the wave telescope technique (Motschmann et al., 1996).

Suppose, that the magnetic field vector $\underline{B}(\underline{x}, \omega_0)$ is measured at $N$ positions $\underline{x}_i$ ($i = 1, \ldots, N$), summarized into the $3N$-dimensional vector $\underline{B}(\omega_0)$. Thus, the determination of the spectral amplitude $\underline{\hat{B}}_0(\underline{k}, \omega_0)$ results in an overdetermined inversion problem. Following Motschmann et al. (1996), Narita (2019) and Toepfer et al. (2020b), the magnetic field model can be rewritten as

$$\underline{B}(\omega_0) = \int \underline{\underline{H}}(\underline{k}) \, \underline{\hat{B}}_0(\underline{k}, \omega_0) \, \mathrm{d}^3 k, \tag{2}$$

where

$$\underline{\underline{H}}(\underline{k}) = \begin{bmatrix} \underline{\underline{I}} e^{i\underline{k} \cdot \underline{x}_1} \\ \vdots \\ \underline{\underline{I}} e^{i\underline{k} \cdot \underline{x}_N} \end{bmatrix} \in \mathbb{R}^{3N \times 3} \tag{3}$$

is the shape matrix and $\underline{\underline{I}} \in \mathbb{R}^{3 \times 3}$ denotes the identity matrix. The magnetic field measurements can be arranged into the data covariance matrix

$$\underline{\underline{M}} = \langle \underline{B}(\omega_0) \circ \underline{B}(\omega_0) \rangle \in \mathbb{R}^{3N \times 3N},$$

where the angular brackets denote the statistical average of the data. The spectrum of the wave can be estimated via

$$P(\underline{k}) = \mathrm{tr} \left\{ \left[ \underline{\underline{H}}^\dagger \underline{\underline{M}}^{-1} \underline{\underline{H}} \right]^{-1} \right\} \tag{4}$$





where the dagger † denotes the Hermitian conjugate and $\mathrm{tr}\left\{\left[\underline{\underline{H}}^\dagger \underline{\underline{M}}^{-1} \underline{\underline{H}}\right]^{-1}\right\}$ is the trace of the matrix $\left[\underline{\underline{H}}^\dagger \underline{\underline{M}}^{-1} \underline{\underline{H}}\right]^{-1}$. The maximum values of $P(\underline{k})$ may be interpreted as the spectrum of the field. If only a finite number of sharp peaks emerges, the magnetic field may be interpreted as a superposition of plane waves with discrete $\underline{k}$-values. As $P(\underline{k})$ is a non-linear function of the vector $\underline{k}$, the whole three-dimensional $\underline{k}$-space needs to be scanned for identifying the peaks (Motschmann et al., 1996).

## 3 Electromagnetic knots

The classical wave telescope technique does not assume any symmetry or relation between different $\underline{k}$-vectors of the spectrum. However, to be able to use electromagnetic knots as a system of basis structures, the geometry of the $\underline{k}$-space needs to be specialized. In this respect, the classical wave telescope technique differs from its extension presented here. The following mathematical derivation of electromagnetic knots is based on Cameron (2018).

### 3.1 Construction of the knots

For the specific evaluation of the integral in Eq. (1), spherical coordinates $(k, \varphi, \theta)$ in the $\underline{k}$-space are introduced

$$\underline{k} = k \begin{pmatrix} \sin\theta\cos\varphi \\ \sin\theta\sin\varphi \\ \cos\theta \end{pmatrix} =: k\,\underline{e}_k, \tag{5}$$

where the corresponding unit vectors are given by

$$\underline{e}_k = \sin\theta\cos\varphi\,\underline{e}_x + \sin\theta\sin\varphi\,\underline{e}_y + \cos\theta\,\underline{e}_z$$
$$\underline{e}_\varphi = -\sin\varphi\,\underline{e}_x + \cos\varphi\,\underline{e}_y$$
$$\underline{e}_\theta = \cos\theta\cos\varphi\,\underline{e}_x + \cos\theta\sin\varphi\,\underline{e}_y - \sin\theta\,\underline{e}_z.$$

The vectors $\underline{e}_x$, $\underline{e}_y$ and $\underline{e}_z$ denote the unit vectors of the cartesian coordinate system.





In this case, the magnetic field in Eq. (1) can be rewritten as

$$
\begin{aligned}
\underline{B}(\underline{x},\omega_0) &= \int \underline{\hat{B}}_0(\underline{k},\omega_0)\, e^{i\underline{k}\cdot\underline{x}}\, \mathrm{d}^3k \\
&= \int \underline{\hat{B}}_0(k,\theta,\varphi,\omega_0) e^{ik\underline{e}_k\cdot\underline{x}}\, \mathrm{d}^3k \\
&= \int_0^\infty \int_0^{2\pi} \int_0^{\pi} \underline{\hat{B}}_0(k,\theta,\varphi,\omega_0) e^{ik\underline{e}_k\cdot\underline{x}} k^2 \sin\theta\, \mathrm{d}\theta\, \mathrm{d}\varphi\, \mathrm{d}k \\
&= \int_0^\infty \left\{ \int_0^{2\pi} \int_0^{\pi} \underline{\hat{B}}_0(k,\theta,\varphi,\omega_0) e^{ik\underline{e}_k\cdot\underline{x}} \sin\theta\, \mathrm{d}\theta\, \mathrm{d}\varphi \right\} k^2\, \mathrm{d}k \\
&= \int_0^\infty \underline{\tilde{b}}(k,\omega_0,\underline{x})\, k^2\, \mathrm{d}k,
\end{aligned}
\tag{6}
$$

where

$$
\underline{\tilde{b}}(k,\omega_0,\underline{x}) = \int_0^{2\pi} \int_0^{\pi} \underline{\hat{B}}_0(k,\theta,\varphi,\omega_0) e^{ik\underline{e}_k\cdot\underline{x}} \sin\theta\, \mathrm{d}\theta\, \mathrm{d}\varphi
\tag{7}
$$

is the spectral amount of the field corresponding to $k$.

Due to Maxwell's equations, the magnetic field (as well as the electric field in the absence of free charge carriers) is solenoidal

$$
\partial_{\underline{x}} \cdot \underline{B}(\underline{x},t) = 0
\tag{8}
$$

such that

$$
\underline{\hat{B}}_0(k,\theta,\varphi,\omega_0) \cdot \underline{e}_k = 0.
\tag{9}
$$

To guarantee the solenoidality of the magnetic field, the ansatz

$$
\underline{\hat{B}}_0(k,\theta,\varphi,\omega_0) = \alpha(k,\varphi,\theta,\omega_0)\, \underline{e}_\varphi + \beta(k,\varphi,\theta,\omega_0)\, \underline{e}_\theta
\tag{10}
$$

is chosen, which results in

$$
\underline{\tilde{b}}(k,\omega_0,\underline{x}) = \int_0^{2\pi} \int_0^{\pi} \left[ \alpha(k,\varphi,\theta,\omega_0)\, \underline{e}_\varphi + \beta(k,\varphi,\theta,\omega_0)\, \underline{e}_\theta \right] e^{ik\underline{e}_k\cdot\underline{x}} \sin\theta\, \mathrm{d}\theta\, \mathrm{d}\varphi,
\tag{11}
$$

as well as

$$
\underline{B}(\underline{x},\omega_0) = \int_0^\infty \int_0^{2\pi} \int_0^{\pi} \left[ \alpha(k,\varphi,\theta,\omega_0)\, \underline{e}_\varphi + \beta(k,\varphi,\theta,\omega_0)\, \underline{e}_\theta \right] e^{ik\underline{e}_k\cdot\underline{x}} \sin\theta\, \mathrm{d}\theta\, \mathrm{d}\varphi\, k^2\, \mathrm{d}k,
\tag{12}
$$





where $\alpha(k,\varphi,\theta,\omega_0)$ and $\beta(k,\varphi,\theta,\omega_0)$ are complex functions of $(k,\varphi,\theta)$ and $\omega_0$. In the following this ansatz is specified by constraining the geometry of the three dimensional $\underline{k}$-space.

Equation (11) represents the spectral amplitude of the magnetic field for a fixed value of $k$. Thus, it is useful to separate the
angular dependency $(\varphi,\theta)$ of the spectral amplitude from the $k$-dependency by choosing the functions $\alpha$ und $\beta$ as

$$\alpha(k,\varphi,\theta,\omega_0) = \alpha'(\varphi,\theta,\omega_0)\frac{B_0}{2\pi}\frac{\mathcal{K}(k)}{k^2} \tag{13}$$

$$\beta(k,\varphi,\theta,\omega_0) = \beta'(\varphi,\theta,\omega_0)\frac{B_0}{2\pi}\frac{\mathcal{K}(k)}{k^2}, \tag{14}$$

where $\mathcal{K}(k)$ is a function of $k$ alone, $\alpha'(\varphi,\theta,\omega_0)$ and $\beta'(\varphi,\theta,\omega_0)$ are complex functions of $(\varphi,\theta,\omega_0)$ and $B_0$ is a real constant.

In this respect, the spectral amplitude (Eq. 11) can be rewritten as

$$\underline{\tilde{b}}(k,\omega_0,\underline{x}) = \frac{B_0}{2\pi}\int\limits_0^{2\pi}\int\limits_0^{\pi}\left[\alpha'(\varphi,\theta)\,\underline{e}_\varphi + \beta'(\varphi,\theta)\,\underline{e}_\theta\right]e^{ik\underline{e}_k\cdot\underline{x}}\sin\theta\,\mathrm{d}\theta\,\mathrm{d}\varphi\,\frac{\mathcal{K}(k)}{k^2}, \tag{15}$$

where the functions $\alpha'(\varphi,\theta)$ and $\beta'(\varphi,\theta)$ weight the summation over the $\underline{k}$-space with respect to the angulars $\varphi$ and $\theta$. Introducing the abbreviation

$$\underline{\tilde{B}}(k,\omega_0,\underline{x}) = \frac{B_0}{2\pi}\int\limits_0^{2\pi}\int\limits_0^{\pi}\left[\alpha'(\varphi,\theta)\,\underline{e}_\varphi + \beta'(\varphi,\theta)\,\underline{e}_\theta\right]e^{ik\underline{e}_k\cdot\underline{x}}\sin\theta\,\mathrm{d}\theta\,\mathrm{d}\varphi \tag{16}$$

provides

$$\underline{\tilde{b}}(k,\omega_0,\underline{x}) = \underline{\tilde{B}}(k,\omega_0,\underline{x})\frac{\mathcal{K}(k)}{k^2}, \tag{17}$$

such that

$$\underline{B}(\underline{x},\omega_0) = \int\limits_0^{\infty}\underline{\tilde{B}}(k,\omega_0,\underline{x})\,\mathcal{K}(k)\,\mathrm{d}k. \tag{18}$$

In the following, the functions $\alpha'(\varphi,\theta)$ and $\beta'(\varphi,\theta)$ are specified, to evaluate the spectral amplitude $\underline{\tilde{B}}(k,\omega_0,\underline{x})$ with regard to electromagnetic knots (Cameron, 2018).

Each spectral amount (corresponding to a fixed $k$-value) of the field may be characterized by a superposition of plane waves with the same amplitude propagating in every direction (independent of $\varphi$ and $\theta$), such that

$$\alpha'(\varphi,\theta) = \alpha_0' = \text{const.} \in \mathbb{C}, \quad \beta'(\varphi,\theta) = \beta_0' = \text{const.} \in \mathbb{C}. \tag{19}$$





In this case, the spectral amplitude results in

$$\underline{\tilde{B}}(k,\omega_0,\underline{x}) = \frac{B_0}{2\pi} \int\limits_0^{2\pi}\int\limits_0^{\pi} \left[\alpha_0'\,\underline{e}_\varphi + \beta_0'\,\underline{e}_\theta\right] e^{ik\underline{e}_k\cdot\underline{x}}\sin\theta\,\mathrm{d}\theta\,\mathrm{d}\varphi, \tag{20}$$

representing a superposition of infinitely many plane waves of the same amplitude with the spectrum

$$\mathcal{S}_k = \left\{\underline{k}\in\mathbb{R}^3\,\big|\,k_x^2 + k_y^2 + k_z^2 = k^2\right\}. \tag{21}$$

Therefore, the distribution in $\underline{k}$-space is completely characterized by the value $k$.

Using the definitions of the unit vectors $\underline{e}_\varphi$ and $\underline{e}_\theta$, the magnetic field can be further expanded into the form

$$\underline{\tilde{B}}(k,\omega_0,\underline{x}) = \frac{B_0}{2\pi}\Bigg\{\alpha_0' \int\limits_0^{2\pi}\int\limits_0^{\pi} \left[-\sin\varphi\,\underline{e}_x + \cos\varphi\,\underline{e}_y\right] e^{ik\underline{e}_k\cdot\underline{x}}\sin\theta\,\mathrm{d}\theta\,\mathrm{d}\varphi$$

$$+ \beta_0' \int\limits_0^{2\pi}\int\limits_0^{\pi} \left[\cos\theta\cos\varphi\,\underline{e}_x + \cos\theta\sin\varphi\,\underline{e}_y - \sin\theta\,\underline{e}_z\right] e^{ik\underline{e}_k\cdot\underline{x}}\sin\theta\,\mathrm{d}\theta\,\mathrm{d}\varphi\Bigg\}$$

$$= \frac{B_0}{2\pi}\Bigg\{\underline{e}_x \int\limits_0^{2\pi}\int\limits_0^{\pi} \left(-\alpha_0'\sin\varphi + \beta_0'\cos\theta\cos\varphi\right) e^{ik\underline{e}_k\cdot\underline{x}}\sin\theta\,\mathrm{d}\theta\,\mathrm{d}\varphi$$

$$+ \underline{e}_y \int\limits_0^{2\pi}\int\limits_0^{\pi} \left(\alpha_0'\cos\varphi + \beta_0'\cos\theta\sin\varphi\right) e^{ik\underline{e}_k\cdot\underline{x}}\sin\theta\,\mathrm{d}\theta\,\mathrm{d}\varphi$$

$$- \beta_0'\,\underline{e}_z \int\limits_0^{2\pi}\int\limits_0^{\pi} \sin^2\theta\,e^{ik\underline{e}_k\cdot\underline{x}}\,\mathrm{d}\theta\,\mathrm{d}\varphi\Bigg\}. \tag{22}$$

For the evaluation of the integrals in Eq. (22) it is useful to introduce a cylindrical coordinate system $(\rho,\phi,z)$ in the position space

$$\underline{x} = \begin{pmatrix} \rho\cos\phi \\ \rho\sin\phi \\ z \end{pmatrix} = \rho\,\underline{e}_\rho + z\,\underline{e}_z, \tag{23}$$

where $\rho = \sqrt{x^2 + y^2}$. The corresponding unit vectors are given by

$$\underline{e}_\rho = \cos\phi\,\underline{e}_x + \sin\phi\,\underline{e}_y$$

$$\underline{e}_\phi = -\sin\phi\,\underline{e}_x + \cos\phi\,\underline{e}_y$$

$$\underline{e}_z = \underline{e}_z.$$





The scalar product of the $\underline{k}$-vector and the position vector results in

$$\underline{e}_k \cdot \underline{x} = \rho\cos\phi\cos\varphi\sin\theta + \rho\sin\phi\sin\varphi\sin\theta + z\cos\theta$$
$$= \rho\sin\theta(\cos\phi\cos\varphi + \sin\phi\sin\varphi) + z\cos\theta. \qquad (24)$$

Using $x = \rho\cos\phi$ and $y = \rho\sin\phi$, provides

$$\underline{e}_k \cdot \underline{x} = \sin\theta(x\cos\varphi + y\sin\varphi) + z\cos\theta \qquad (25)$$

For the further evaluation of the integrals in each component of Eq. (22), the abbreviations

$$\eta_1(\theta) := k_0 x \sin\theta \quad \text{and} \quad \eta_2(\theta) := k_0 y \sin\theta$$

are introduced. By means of these preparations, the $\varphi$-integration can be solved analytically, delivering the Bessel functions of the first kind

$$\mathcal{J}_n(x) = \frac{1}{2\pi}\int\limits_{-\pi}^{\pi} e^{i(n\tau - x\sin\tau)}\,\mathrm{d}\tau.$$

The detailed evaluation of the integrals can be found in the appendix, resulting in

$$\underline{\tilde{B}}(k,\omega_0,\underline{x}) = \mathrm{Re}\left\{B_0\left[i\alpha_0' f\,\underline{e}_\phi + \beta_0'\left(g\,\underline{e}_\rho + h\,\underline{e}_z\right)\right]\right\}, \qquad (26)$$

and

$$\underline{B}(\underline{x},\omega_0) = \int\limits_0^\infty \underline{\tilde{B}}(k,\omega_0,\underline{x})\,\mathcal{K}(k)\,\mathrm{d}k = \int\limits_0^\infty \mathrm{Re}\left\{B_0\left[i\alpha_0' f(\underline{x},k)\,\underline{e}_\phi + \beta_0'\left(g(\underline{x},k)\,\underline{e}_\rho + h(\underline{x},k)\,\underline{e}_z\right)\right]\right\}\mathcal{K}(k)\,\mathrm{d}k, \qquad (27)$$

where

$$f(\underline{x},k) := \int\limits_0^\pi \sin\theta\,\cos(kz\cos\theta)\,\mathcal{J}_1(k\rho\sin\theta)\,\mathrm{d}\theta$$

$$g(\underline{x},k) := -\int\limits_0^\pi \sin\theta\cos\theta\,\sin(kz\cos\theta)\,\mathcal{J}_1(k\rho\sin\theta)\,\mathrm{d}\theta$$

and

$$h(\underline{x},k) := -\int\limits_0^\pi \sin^2\theta\,\cos(kz\cos\theta)\,\mathcal{J}_0(k\rho\sin\theta)\,\mathrm{d}\theta.$$

The complex constants $\alpha_0'$ and $\beta_0'$ are the free parameters of the magnetic field in Eq. (26) and can be chosen independently of

each other. The first part of the field

$$\underline{\tilde{B}}_r(k,\omega_0,\underline{x}) := \mathrm{Re}\left\{B_0 i\alpha_0' f\,\underline{e}_\phi\right\} \quad \text{or} \quad \underline{B}_r(\underline{x},\omega_0) = \int\limits_0^\infty \mathrm{Re}\left\{B_0 i\alpha_0' f\,\underline{e}_\phi\right\}\mathcal{K}(k)\,\mathrm{d}k, \qquad (28)$$





that corresponds to the expansion coefficient $\alpha'_0$, is called the magnetic ring (cf. Fig. 1a). The second part

$$\underline{\tilde{B}}_g(k,\omega_0,\underline{x}) := \mathrm{Re}\left\{B_0\beta'_0\left(g\,\underline{e}_\rho + h\,\underline{e}_z\right)\right\} \quad \text{or} \quad \underline{B}_g(\underline{x},\omega_0) = \int_0^\infty \mathrm{Re}\left\{B_0\beta'_0\left(g\,\underline{e}_\rho + h\,\underline{e}_z\right)\right\}\mathcal{K}(k)\,\mathrm{d}k, \tag{29}$$

corresponding to the expansion coefficient $\beta'_0$, is the magnetic globule (cf. Fig. 1b). Thus, the magnetic ring and the magnetic

globule can be interpreted as a set of basis functions for localized, divergence-free structures.



## 3.2 Electric field

The electric and the magnetic field, respectively, are connected via Ampère's law. Under the absence of Ohmic currents, Ampère's law reduces to

$$\partial_{\underline{x}} \times \underline{B}(\underline{x},t) = \frac{1}{c_{ph}^2} \partial_t \underline{E}(\underline{x},t), \tag{30}$$

where $c_{ph}$ is the phase velocity. Fourier transformation provides

$$i\underline{k} \times \underline{B}(\underline{x},\omega) = -i\frac{\omega}{c_{ph}^2} \underline{E}(\underline{x},\omega). \tag{31}$$

Using $\underline{k} = k\,\underline{e}_k$, yields

$$\underline{e}_k \times \underline{B}(\underline{x},\omega) = -\frac{\omega}{kc_{ph}^2} \underline{E}(\underline{x},\omega), \tag{32}$$

such that

$$\underline{E}(\underline{x},\omega) = -\frac{kc_{ph}^2}{\omega} \underline{e}_k \times \underline{B}(\underline{x},\omega). \tag{33}$$

Ampère's law is valid for every $\underline{k}$-vector that contributes to the spectrum of the field, yielding the ansatz

$$\underline{\tilde{E}}(k,\omega,\underline{x}) = -\frac{kc_{ph}^2}{\omega} \underline{e}_k \times \underline{\tilde{B}}(k,\omega,\underline{x}). \tag{34}$$

Using

$$\underline{e}_k \times \underline{e}_\varphi = -\underline{e}_\theta \quad \text{and} \quad \underline{e}_k \times \underline{e}_\theta = \underline{e}_\varphi,$$

delivers

$$\underline{\tilde{E}}(k,\omega,\underline{x}) = -\frac{kc_{ph}^2}{\omega} \frac{B_0}{2\pi} \int\limits_0^{2\pi}\int\limits_0^{\pi} \left[\beta_0'\,\underline{e}_\varphi - \alpha_0'\,\underline{e}_\theta\right] e^{ik\underline{e}_k\cdot\underline{x}} \sin\theta\,\mathrm{d}\theta\,\mathrm{d}\varphi \tag{35}$$

such that the real part can be expressed as

$$\underline{\tilde{E}}(k,\omega,\underline{x}) = -\frac{kc_{ph}^2}{\omega} \mathrm{Re}\left\{ B_0 \left[i\beta_0' f(\underline{x},k)\,\underline{e}_\phi - \alpha_0' \left(g(\underline{x},k)\,\underline{e}_\rho + h(\underline{x},k)\,\underline{e}_z\right)\right] \right\}. \tag{36}$$

Thus, the electric field is given by

$$\underline{E}(\underline{x},\omega_0) = -\int\limits_0^\infty \frac{kc_{ph}^2}{\omega_0} \mathrm{Re}\left\{ B_0 \left[i\beta_0' f(\underline{x},k)\,\underline{e}_\phi - \alpha_0' \left(g(\underline{x},k)\,\underline{e}_\rho + h(\underline{x},k)\,\underline{e}_z\right)\right] \right\} \mathcal{K}(k)\,\mathrm{d}k. \tag{37}$$





### 3.3 Electric current density

When Ohmic currents $\underline{j}(\underline{x},t) \neq 0$ are present, Ampère's law can be written as

$$\partial_{\underline{x}} \times \underline{B}(\underline{x},t) = \mu_0 \underline{j}(\underline{x},t), \tag{38}$$

under the assumption of stationarity or if the displacement current is negligible. Again, Fourier transformation provides

$$i\underline{k} \times \underline{B}(\underline{x},\omega) = \mu_0 \underline{j}(\underline{x},\omega), \tag{39}$$

such that

$$\tilde{\underline{j}}(k,\omega,\underline{x}) = i\frac{k}{\mu_0} \underline{e}_k \times \underline{\tilde{B}}(k,\omega,\underline{x}) \tag{40}$$

In analogy to the electric field, the current density can be calculated via

$$\underline{j}(\underline{x},\omega_0) = \int\limits_0^\infty \frac{k}{\mu_0} \mathrm{Re}\left\{ iB_0 \left[ i\beta_0' f(\underline{x},k)\,\underline{e}_\phi - \alpha_0' \left( g(\underline{x},k)\,\underline{e}_\rho + h(\underline{x},k)\,\underline{e}_z \right) \right] \right\} \mathcal{K}(k)\,\mathrm{d}k. \tag{41}$$

Thus, the current density of the magnetic ring follows the topology of a globule and vice versa.

### 3.4 Spatially distributed knot structures

Within the derivation of the knot structures, the magnetic ring and the magnetic globule are defined with respect to the same origin of the cylindrical coordinate system $(\rho,\phi,z)$. The resulting structures are also known as (electro-)magnetic disturbances of the first kind (Cameron, 2018). However, in general the structures can be defined with respect to different (local) coordinate
systems, spanned by the local unit vectors $(\underline{e}_{\rho_q}, \underline{e}_{\phi_q}, \underline{e}_{z_q})$, where $q = 1, \ldots, Q$, with different origins $\underline{\mathcal{O}}_q$. The resulting structures

$$\underline{B}(\underline{x},\omega_0) = \int\limits_0^\infty \mathrm{Re}\left\{ B_0 \sum_{q=1}^Q \left[ i\alpha_{0q}' f(\underline{x}_q,k)\,\underline{e}_{\phi_q} + \beta_{0q}' \left( g(\underline{x}_q,k)\,\underline{e}_{\rho_q} + h(\underline{x}_q,k)\,\underline{e}_{z_q} \right) \right] \right\} \mathcal{K}(k)\,\mathrm{d}k \tag{42}$$

with

$$\underline{x}_q = \underline{\mathcal{O}}_q + \rho_q \underline{e}_{\rho_q} + z_q \underline{e}_{z_q}$$

are a superposition of $Q$ translated and/or rotated (electro-)magnetic disturbances of the first kind (cf. Fig. 1c) and are also called (electro-)magnetic disturbances of the second kind (Cameron, 2018). The field is characterized by $8Q$ free parameters, i.e., the expansion coefficients $\alpha_{0q}'$ and $\beta_{0q}'$, the origins $\underline{\mathcal{O}}_q$ as well as the orientation of the local coordinate system that can be described for example via Euler angles (Cameron, 2018).





### 3.5 Discussion of the knot structures

Within the above presented derivation, the spectral distribution of the field with respect to $k$ is controlled by the function $\mathcal{K}(k)$. Electromagnetic knots, as originally described by Cameron (2018) are superpositions of infinitely many, monochromatic plane waves, i.e., $\mathcal{K}(k) = \delta(k - k_0)$, with the same amplitude, propagating in every direction with the spectrum

$$\mathcal{S}_{k_0} = \left\{ \underline{k} \in \mathbb{R}^3 \,|\, k_x^2 + k_y^2 + k_z^2 = k_0^2 \right\}. \tag{43}$$

In contrast to single plane waves, knots are localized structures, similar to wave packages. The localization of the structures
results from the spatial distribution of the wave phases

$$\mathcal{F}(\theta, \varphi) := \underline{e}_k \cdot \underline{x} = \sin\theta \left( x\cos\varphi + y\sin\varphi \right) + z\cos\theta. \tag{44}$$

Thus, the knots are a superposition of plane waves with different phases $\mathcal{F}(\theta, \varphi)$ at all points in space despite its central point. At the origin of the structure $(x = y = z = 0)$ the phases of the waves are all equal: $\mathcal{F}(\theta, \varphi) = 0$, resulting in a constructive interference with a maximum amplitude at the central point. The scale size of the knot is determined by $k_0$, representing a set
of infinitely many $\underline{k}$-vectors with the same length. The superposition of the plane waves is schematically illustrated in Fig. (2)

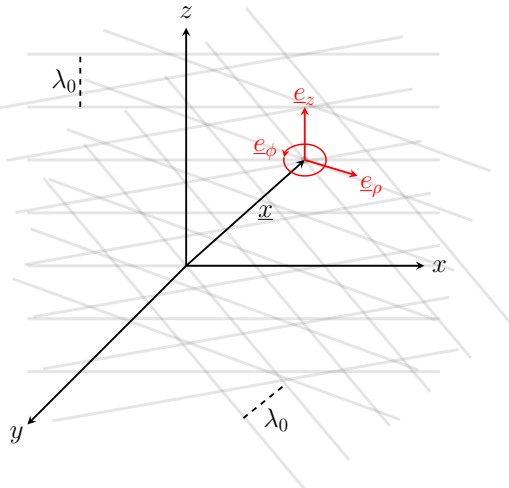

**Figure 2.** Illustration of superposed, monochromatic plane wave fronts (gray lines) with the wave length $\lambda_0 = 2\pi/k_0$. The knots are localized in the origin of the red coordinate system spanned by the vectors $\underline{e}_\rho$, $\underline{e}_\phi$ and $\underline{e}_z$.

Equation (27) represents the magnetic with respect to the position vector $\underline{x}$ an the frequency $\omega_0$. However, the spatial structure of the field can also directly be analyzed from the measurement data $\underline{B}(\underline{x}, t)$ evaluated at different time steps $t$ and thus, no Fourier transform with respect to time is required.





## 4 Extension of the wave telescope

Following this short derivation and discussion of the electromagnetic knots, the knot model needs to be reformulated in terms of the wave telescope technique for estimating the spectrum of the knots.

### 4.1 Reformulation of the model

After performing the temporal Fourier transform, the magnetic field (Eq. 27), measured at the position $\underline{x}_i$, $i = 1, \ldots, N$, can be rewritten as

$$\underline{B}(\underline{x}_i, \omega_0) = \int_0^\infty \underline{\tilde{B}}(k, \omega_0, \underline{x}_i) \mathcal{K}(k) \, \mathrm{d}k = \int_0^\infty \mathrm{Re}\left\{ \underline{\underline{H}}_i(k) \begin{pmatrix} \hat{B}_0(k, \omega_0) i\alpha_0' \\ \hat{B}_0(k, \omega_0) \beta_0' \end{pmatrix} \right\} \mathcal{K}(k) \, \mathrm{d}k, \tag{45}$$

where

$$\underline{\underline{H}}_i(k) = \begin{bmatrix} -f(\underline{x}_i, k) \sin\phi & g(\underline{x}_i, k) \cos\phi \\ f(\underline{x}_i, k) \cos\phi & g(\underline{x}_i, k) \sin\phi \\ 0 & h(\underline{x}_i, k) \end{bmatrix} \tag{46}$$

is the corresponding shape matrix of the position $\underline{x}_i$. Summarizing the measurements into a $3N$-dimensional vector $\underline{B}(\omega_0)$, the magnetic field can be rearranged as

$$\underline{B}(\omega_0) = \int_0^\infty \mathrm{Re}\left\{ \underline{\underline{H}}(k) \begin{pmatrix} \hat{B}_0(k, \omega_0) i\alpha_0' \\ \hat{B}_0(k, \omega_0) \beta_0' \end{pmatrix} \right\} \mathcal{K}(k) \, \mathrm{d}k, \tag{47}$$

where

$$\underline{\underline{H}}(k) := \begin{bmatrix} \underline{\underline{H}}_1(k) \\ \vdots \\ \underline{\underline{H}}_N(k) \end{bmatrix} \in \mathbb{R}^{3N \times 2}. \tag{48}$$

Again, the determination of the amplitudes $\alpha_0' \hat{B}_0(k, \omega_0)$ and $\beta_0' \hat{B}_0(k, \omega_0)$ results in an overdetermined inversion problem. In analogy to the classical wave telescope technique, the spectrum of the ring can be estimated via

$$P(k) = \mathrm{tr}\left\{ \left[ \underline{\underline{H}}^\dagger \underline{\underline{M}}^{-1} \underline{\underline{H}} \right]^{-1} \right\}. \tag{49}$$

Since $P(k)$ is a non-linear function of $k$, the whole $k$-space has to be scanned for estimating the spectrum of the field (Motschmann et al., 1996).

Solely considering the magnetic ring (Eq. 28), the shape matrix transfers onto the shape vector (Narita, 2019, cf.)

$$\underline{h}_r(k) := f(\underline{x}, k) \underline{e}_\phi = \begin{pmatrix} -f(\underline{x}, k) \sin\phi \\ f(\underline{x}, k) \cos\phi \\ 0 \end{pmatrix}. \tag{50}$$



In this case, the spectrum of the ring can be estimated via

$$P_r(k) = \frac{1}{\underline{h}_r^\dagger(k)\underline{\underline{M}}^{-1}\underline{h}_r(k)}. \tag{51}$$

## 4.2 Application to plasmoids

For the first application of electromagnetic knots in the context of magnetospheric structures, we consider the modeling of plasmoids via a magnetic ring (Zhang et al., 2013, cf.). Plasmoids are a consequence of magnetic reconnection in the far tail region of a planetary magnetosphere triggered by the Dungey cycle (McPherron, 1995, e.g.). The structures are characterized by a magnetic ring along the neutral sheet line with a length scale of the order of the solar wind's obstacle (McPherron, 1995; Zong et al., 2004, e.g.).


We model the magnetic field in the tail region by superposing a stationary magnetic ring ($\alpha_0' = -i$, Eq. 28)

$$\underline{B}_r(\underline{x}, \omega_0 = 0) = \int_0^\infty \mathrm{Re}\left\{B_0 i\alpha_0' f(\underline{x}, k)\,\underline{e}_\phi\right\}\delta(k - k_0)\,\mathrm{d}k = \mathrm{Re}\left\{B_0 i\alpha_0' f(\underline{x}, k_0)\,\underline{e}_\phi\right\} = B_0\,\underline{h}_r(k_0) \tag{52}$$

composed of monochromatic plane waves, representing the plasmoid, with the field generated by the neutral sheet current (Harris neutral sheet, Harris (1962)), such that

$$\underline{B}(\underline{x}) = \underline{B}_r(\underline{x}, \omega_0 = 0) - B_s \tanh\left(\frac{y}{L}\right)\underline{e}_x = B_0\,\underline{h}_r(\underline{x}, k_0) - B_s \tanh\left(\frac{y}{L}\right)\underline{e}_x, \tag{53}$$

where the $x$-axis points towards the night side magnetosphere, the $y$-axis points from the southern geographic pole to the northern geographic pole and the $z$-axis completes the right-handed system. Thus, we model the plasmoid as a two-dimensional structure in the $x$-$y$-plane (Zhang et al., 2013, cf.). The value $B_0$ represents an arbitrary chosen background amplitude, $B_s = 0.3\,B_0$ and the length scale of the current sheet is chosen to $L = 10^{-3}\,R_\mathrm{E}$, where $R_\mathrm{E}$ is the planetary radius, e.g., the terres-
trial radius. The characteristic length scale of the plasmoid is chosen to $\lambda_0 = 1.5\,R_\mathrm{E}$, corresponding to $k_0 = 2\pi/\lambda_0 \approx 4.19\,R_\mathrm{E}^{-1}$.

The resulting magnetic field data are evaluated at $N = 7$ synthetically generated spacecraft positions, representing an HelioSwarm-like configuration (Klein and Spence, 2021). As plasmoids are highly dynamical, travelling structures, the measurement positions are shifted along the $x$-axis with respect to the origin of the plasmoid (left, mean, right), representing
different time steps. The length scale $\lambda_0$ (or equivalently $k_0$) of the plasmoid is estimated from the virtual spacecraft data via Eq. (51). The resulting field geometry (blue arrows) and the measurement positions (red dots) as well es the corresponding spectra are illustrated in Fig. (3).

When the measurement positions are distributed around the origin of the plasmoid (mean), the implemented value of $k_0$ can
be reconstructed with high precision from the data. In the other cases, the spatial length scale is slightly overestimated and





the relative error results in about $6\%$ (left) and $4\%$ (right), respectively. Thus, the wave telescope technique is capable of (1) seperating the plasmoid from the neutral sheet part and (2) of estimating the characteristic length scale of the plasmoid from a limited number of measurement positions.

In analogy to the classical wave telescope technique, the accuracy of the reconstruction depends of the relation between the plasmoid's length scale $\lambda_0$ and the mean distance $d$ between the spacecraft positions (Narita et al., 2022, e.g.). For example, if $d \ll \lambda_0$, the measurement positions do not properly cover the spatial extend of the plasmoid, resulting in ambiguities within the reconstruction procedure. In the case of $d \gg \lambda_0$, the magnetic field structure of the plasmoid is not detectable. Thus, the mean distance between the spacecraft positions has to be of the order of the plasmoid's spatial scale $d \sim \lambda_0$, which will be realized
by the configuration of the planned HelioSwarm multiscale mission.





 

**Figure 3.** Reconstructed spectrum $P_r(k)$ resulting from different measurement positions (red dots) with respect to the origin of the plasmoid. The length scale of the plasmoid is chosen to $k_0 \approx 4.19\,R_{\mathrm{E}}^{-1}$.





### 4.3 Further applications

The above presented application of electromagnetic knots indicates the potential of the representation. Spatially distributed electromagnetic knots as described by Cameron (2018) enable the modeling of more complex structures, provide generalized spectral information, and open the door for further applications, delivering an alternative interpretation of magnetospheric

structures. For example, the magnetic field configuration resulting from a field-aligned current can be modelled as a superposition of magnetic rings stacked on top of each other. Due to Ampère's law, the corresponding current density is given as a superposition of globules. Thus, the inner structure of field-aligned currents can be analyzed directly from the magnetic field measurements (Toepfer et al., 2021, cf.). Also, the current system of Alfvén wings can be described as a superposition of rings (Vernisse et al., 2018, e.g.), so that the corresponding magnetic field topology follows the structure of superposed globules.

Furthermore, field line resonances (Glassmeier et al., 1999; Plaschke et al., 2008) may be described as a special superposition of magnetic rings.



## 5 Conclusions

Electromagnetic knots are a superposition of infinitely many, monochromatic plane waves with a spherical symmetric spectrum and represent an exact solution of Maxwell's equation. The resulting basis elements, i.e., the electromagnetic ring and the globule, form a basis set for localized, divergence-free spatial structures. For this reason, the concept of electromagnetic knots opens the door for a completely new description and interpretation of spatial structures in planetary magnetospheres.

The classification of spatial structures evaluated at a limited number of measurement points describes an overdetermined inversion problem. The wave telescope technique serves as a robust data analysis tool for the global interpretation of spacecraft measurements in terms of expected physical structures. By reformulating the formalism of electromagnetic knots in terms of the wave telescope technique, we extended the zoo of spatial structures that can be analyzed by the method. In this sense, the present study can be interpreted as a generalization of the wave telescope technique to a structure telescope technique.

For a first validation, the concept of electromagnetic knots has been applied to the modeling of a plasmoid. Using a HelioSwarm-like satellite configuration the wave telescope technique is capable of separating the plasmoid, modelled as a magnetic ring, from the field generated by the neutral sheet current and enables the estimation of the length scale of the ring. Thus, the presented extension of the wave telescope technique serves as a new data analysis tool for multi-spacecraft missions, such as the planned HelioSwarm mission. However, the application of electromagnetic knots for characterizing further structures, such as field-aligned currents or Alfvén wings should be analyzed in future studies. In general, we conclude that the modified wave telescope technique outlined here, bears the potential for a new representation and physical description of complex spatial structures existing in space plasmas.





## Appendix A: Evaluation of the integrals

The $x$-**component** of the magnetic field in Eq. (22) can be rewritten as

$$\int_0^{2\pi}\int_0^{\pi} \left(-\alpha_0'\sin\varphi + \beta_0'\cos\theta\cos\varphi\right) e^{ik\underline{e}_k\cdot\underline{x}} \sin\theta\,\mathrm{d}\theta\,\mathrm{d}\varphi$$

$$= \int_0^{\pi} \sin\theta\, e^{ikz\cos\theta} \int_0^{2\pi} \left(-\alpha_0'\sin\varphi + \beta_0'\cos\theta\cos\varphi\right) e^{i(kx\cos\varphi + ky\sin\varphi)\sin\theta}\,\mathrm{d}\varphi\,\mathrm{d}\theta$$

$$= -\alpha_0' \int_0^{\pi} \sin\theta\, e^{ikz\cos\theta} \int_0^{2\pi} \sin\varphi\, e^{i\eta_1\cos\varphi} e^{i\eta_2\sin\varphi}\,\mathrm{d}\varphi\,\mathrm{d}\theta$$

$$+ \beta_0' \int_0^{\pi} \sin\theta\cos\theta\, e^{ikz\cos\theta} \int_0^{2\pi} \cos\varphi\, e^{i\eta_1\cos\varphi} e^{i\eta_2\sin\varphi}\,\mathrm{d}\varphi\,\mathrm{d}\theta$$

$$= -\alpha_0' \int_0^{\pi} \sin\theta\, e^{ikz\cos\theta} I_1(\theta)\,\mathrm{d}\theta + \beta_0' \int_0^{\pi} \sin\theta\cos\theta\, e^{ikz\cos\theta} I_2(\theta)\,\mathrm{d}\theta \tag{A1}$$

where

$$I_1(\theta) := \int_0^{2\pi} \sin\varphi\, e^{i\eta_1\cos\varphi} e^{i\eta_2\sin\varphi}\,\mathrm{d}\varphi \tag{A2}$$

and

$$I_2(\theta) := \int_0^{2\pi} \cos\varphi\, e^{i\eta_1\cos\varphi} e^{i\eta_2\sin\varphi}\,\mathrm{d}\varphi. \tag{A3}$$

Analogously, the $y$-**component** in Eq. (22) results in

$$\int_0^{2\pi}\int_0^{\pi} \left(\alpha_0'\cos\varphi + \beta_0'\cos\theta\sin\varphi\right) e^{ik\underline{e}_k\cdot\underline{x}} \sin\theta\,\mathrm{d}\theta\,\mathrm{d}\varphi$$

$$= \int_0^{\pi} \sin\theta\, e^{ikz\cos\theta} \int_0^{2\pi} \left(\alpha_0'\cos\varphi + \beta_0'\cos\theta\sin\varphi\right) e^{i\eta_1\cos\varphi} e^{i\eta_2\sin\varphi}\,\mathrm{d}\varphi\,\mathrm{d}\theta$$

$$= \alpha_0' \int_0^{\pi} \sin\theta\, e^{ikz\cos\theta} \int_0^{2\pi} \cos\varphi\, e^{i\eta_1\cos\varphi} e^{i\eta_2\sin\varphi}\,\mathrm{d}\varphi\,\mathrm{d}\theta$$

$$+ \beta_0' \int_0^{\pi} \sin\theta\cos\theta\, e^{ikz\cos\theta} \int_0^{2\pi} \sin\varphi\, e^{i\eta_1\cos\varphi} e^{i\eta_2\sin\varphi}\,\mathrm{d}\varphi\,\mathrm{d}\theta$$

$$= \alpha_0' \int_0^{\pi} \sin\theta\, e^{ikz\cos\theta} I_2(\theta)\,\mathrm{d}\theta + \beta_0' \int_0^{\pi} \sin\theta\cos\theta\, e^{ikz\cos\theta} I_1(\theta)\,\mathrm{d}\theta \tag{A4}$$



as well as

$$-\beta_0' \int_0^{2\pi} \int_0^\pi \sin^2\theta \, e^{ik\underline{e}_k \cdot \underline{x}} \, d\theta \, d\varphi = -\beta_0' \int_0^\pi \sin^2\theta \, e^{ikz\cos\theta} \int_0^{2\pi} e^{i\eta_1\cos\varphi} \, e^{i\eta_2\sin\varphi} \, d\varphi \, d\theta$$

$$= -\beta_0' \int_0^\pi \sin^2\theta \, e^{ikz\cos\theta} I_3(\theta) \, d\theta \tag{A5}$$

for the $z$-**component** in Eq. (22), where

$$I_3(\theta) := \int_0^{2\pi} e^{i\eta_1\cos\varphi} \, e^{i\eta_2\sin\varphi} \, d\varphi. \tag{A6}$$

So far the magnetic field is given by

$$\underline{\tilde{B}}(k,\omega_0,\underline{x}) = \frac{B_0}{2\pi} \Bigg\{ \left[ -\alpha_0' \int_0^\pi \sin\theta \, e^{ikz\cos\theta} I_1(\theta) \, d\theta + \beta_0' \int_0^\pi \sin\theta\cos\theta \, e^{ikz\cos\theta} I_2(\theta) \, d\theta \right] \underline{e}_x$$

$$+ \left[ \alpha_0' \int_0^\pi \sin\theta \, e^{ikz\cos\theta} I_2(\theta) \, d\theta + \beta_0' \int_0^\pi \sin\theta\cos\theta \, e^{ikz\cos\theta} I_1(\theta) \, d\theta \right] \underline{e}_y$$

$$- \beta_0' \int_0^\pi \sin^2\theta \, e^{ikz\cos\theta} I_3(\theta) \, d\theta \, \underline{e}_z \Bigg\}$$

$$= \frac{B_0}{2\pi} \Bigg\{ \alpha_0' \left[ -\int_0^\pi \sin\theta \, e^{ikz\cos\theta} I_1(\theta) \, d\theta \, \underline{e}_x + \int_0^\pi \sin\theta \, e^{ikz\cos\theta} I_2(\theta) \, d\theta \, \underline{e}_y \right]$$

$$+ \beta_0' \left[ \int_0^\pi \sin\theta\cos\theta \, e^{ikz\cos\theta} I_2(\theta) \, d\theta \, \underline{e}_x + \int_0^\pi \sin\theta\cos\theta \, e^{ikz\cos\theta} I_1(\theta) \, d\theta \, \underline{e}_y \right]$$

$$- \beta_0' \int_0^\pi \sin^2\theta \, e^{ikz\cos\theta} I_3(\theta) \, d\theta \, \underline{e}_z \Bigg\}. \tag{A7}$$

At least, the remaining integrals $I_1(\theta)$, $I_2(\theta)$ and $I_3(\theta)$ have to be evaluated.

For the evaluation of the integral

$$I_3(\theta) = \int_0^{2\pi} e^{i\eta_1\cos\varphi} \, e^{i\eta_2\sin\varphi} \, d\varphi = \int_0^{2\pi} e^{i(\eta_1\cos\varphi + \eta_2\sin\varphi)} \, d\varphi \tag{A8}$$

we define

$$\tan\gamma_0 := \frac{\eta_2}{\eta_1},$$





such that

$$\sin\gamma_0 = \frac{\eta_2}{\sqrt{\eta_1^2 + \eta_2^2}}$$

as well as

$$\cos\gamma_0 = \frac{\eta_1}{\sqrt{\eta_1^2 + \eta_2^2}}.$$

Thus, the argument of the complex exponential can be rewritten as

$$
\begin{aligned}
\eta_1\cos\varphi + \eta_2\sin\varphi &= \sqrt{\eta_1^2 + \eta_2^2}\left(\frac{\eta_1}{\sqrt{\eta_1^2 + \eta_2^2}}\cos\varphi + \frac{\eta_2}{\sqrt{\eta_1^2 + \eta_2^2}}\sin\varphi\right) \\
&= \sqrt{\eta_1^2 + \eta_2^2}\left(\cos\gamma_0\cos\varphi + \sin\gamma_0\sin\varphi\right) \\
&= \sqrt{\eta_1^2 + \eta_2^2}\sin(\varphi + \gamma_0).
\end{aligned}
\tag{A9}
$$

Substituting $\tau := \varphi + \gamma_0 + \pi$ and using $\sin(\varphi + \gamma_0) = \sin(\tau - \pi) = -\sin\tau$, delivers

$$I_3(\theta) = \int_{-\pi+\gamma_0}^{\pi+\gamma_0} e^{-i\sqrt{\eta_1^2 + \eta_2^2}\sin\tau}\,\mathrm{d}\tau. \tag{A10}$$

As the integrand is a $2\pi$-periodic function, the integral is independent of $\gamma_0$, so that

$$I_3(\theta) = \int_{-\pi+\gamma_0}^{\pi+\gamma_0} e^{-i\sqrt{\eta_1^2 + \eta_2^2}\sin\tau}\,\mathrm{d}\tau = \int_{-\pi}^{\pi} e^{-i\sqrt{\eta_1^2 + \eta_2^2}\sin\tau}\,\mathrm{d}\tau. \tag{A11}$$

Making use of the definition of the Bessel functions of the first kind

$$\mathcal{J}_n(x) = \frac{1}{2\pi}\int_{-\pi}^{\pi} e^{i(n\tau - x\sin\tau)}\,\mathrm{d}\tau \tag{}$$

yields

$$I_3(\theta) = 2\pi\mathcal{J}_0\left(\sqrt{\eta_1^2 + \eta_2^2}\right) = 2\pi\mathcal{J}_0(k\rho\sin\theta). \tag{A12}$$

The integral

$$I_1(\theta) = \int_0^{2\pi} \sin\varphi\, e^{i\eta_1\cos\varphi}\, e^{i\eta_2\sin\varphi}\,\mathrm{d}\varphi \tag{A13}$$

can be evaluated using the identity

$$\partial_{\eta_2} e^{i\eta_2\sin\varphi} = i\sin\varphi\, e^{i\eta_2\sin\varphi} \tag{A14}$$





so that

$$\sin\varphi\, e^{i\eta_2\sin\varphi} = -i\partial_{\eta_2} e^{i\eta_2\sin\varphi}$$

and results in

$$I_1(\theta) = -i\partial_{\eta_2}\int_0^{2\pi} e^{i\eta_1\cos\varphi}\, e^{i\eta_2\sin\varphi}\, \mathrm{d}\varphi$$

$$= -2\pi i\partial_{\eta_2}\mathcal{J}_0\left(\sqrt{\eta_1^2+\eta_2^2}\right)$$

$$= -2\pi i\frac{\eta_2}{\sqrt{\eta_1^2+\eta_2^2}}\mathcal{J}_0'\left(\sqrt{\eta_1^2+\eta_2^2}\right)$$

$$= -2\pi i\sin\phi\,\mathcal{J}_0'\left(\sqrt{\eta_1^2+\eta_2^2}\right) \tag{A15}$$

by means of Eq. (A12). Using

$$\mathcal{J}_0'(x) = -\mathcal{J}_1(x), \tag{A16}$$

delivers

$$I_1(\theta) = 2\pi i\sin\phi\,\mathcal{J}_1(k\rho\sin\theta). \tag{A17}$$

Analogously, the integral

$$I_2(\theta) = \int_0^{2\pi}\cos\varphi\, e^{i\eta_1\cos\varphi}\, e^{i\eta_2\sin\varphi}\, \mathrm{d}\varphi \tag{A18}$$

can be evaluated using

$$\partial_{\eta_1} e^{i\eta_1\cos\varphi} = i\cos\varphi\, e^{i\eta_1\sin\varphi} \tag{A19}$$

such that

$$\cos\varphi\, e^{i\eta_1\cos\varphi} = -i\partial_{\eta_1} e^{i\eta_1\cos\varphi} \tag{A20}$$

and results in

$$I_2(\theta) = -i\partial_{\eta_1}\int_0^{2\pi} e^{i\eta_1\cos\varphi}\, e^{i\eta_2\sin\varphi}\, \mathrm{d}\varphi$$

$$= -2\pi i\partial_{\eta_1}\mathcal{J}_0\left(\sqrt{\eta_1^2+\eta_2^2}\right)$$

$$= -2\pi i\frac{\eta_1}{\sqrt{\eta_1^2+\eta_2^2}}\mathcal{J}_0'\left(\sqrt{\eta_1^2+\eta_2^2}\right)$$

$$= 2\pi i\frac{\eta_1}{\sqrt{\eta_1^2+\eta_2^2}}\mathcal{J}_1\left(\sqrt{\eta_1^2+\eta_2^2}\right)$$

$$= 2\pi i\cos\phi\,\mathcal{J}_1(k\rho\sin\theta) \tag{A21}$$





Therefore, the magnetic field is given by

$$
\begin{aligned}
\underline{\tilde{B}}(k,\omega_0,\underline{x}) = B_0\Big\{ & i\alpha_0'\Big[ -\sin\phi \int_0^\pi \sin\theta\,e^{ikz\cos\theta}\,\mathcal{J}_1(k\rho\sin\theta)\,\mathrm{d}\theta\,\underline{e}_x \\
& + \cos\phi \int_0^\pi \sin\theta\,e^{ikz\cos\theta}\,\mathcal{J}_1(k\rho\sin\theta)\,\mathrm{d}\theta\,\underline{e}_y\Big] \\
& + \beta_0'\Big[ i\cos\phi \int_0^\pi \sin\theta\cos\theta\,e^{ikz\cos\theta}\,\mathcal{J}_1(k\rho\sin\theta)\,\mathrm{d}\theta\,\underline{e}_x \\
& + i\sin\phi \int_0^\pi \sin\theta\cos\theta\,e^{ikz\cos\theta}\,\mathcal{J}_1(k\rho\sin\theta)\,\mathrm{d}\theta\,\underline{e}_y\Big] \\
& - \beta_0' \int_0^\pi \sin^2\theta\,e^{ikz\cos\theta}\,\mathcal{J}_0(k\rho\sin\theta)\,\mathrm{d}\theta\,\underline{e}_z\Big\} \\
= B_0\Big\{ & i\alpha_0' \int_0^\pi \sin\theta\,e^{ikz\cos\theta}\,\mathcal{J}_1(k\rho\sin\theta)\,\mathrm{d}\theta\,\underline{e}_\phi \\
& + \beta_0' i \int_0^\pi \sin\theta\cos\theta\,e^{ikz\cos\theta}\,\mathcal{J}_1(k\rho\sin\theta)\,\mathrm{d}\theta\,\underline{e}_\rho \\
& - \beta_0' \int_0^\pi \sin^2\theta\,e^{ikz\cos\theta}\,\mathcal{J}_0(k\rho\sin\theta)\,\mathrm{d}\theta\,\underline{e}_z\Big\}.
\end{aligned}
\tag{A22}
$$

The remaining integrals can be expanded into the form

$$
\begin{aligned}
\Phi :=& \int_0^\pi \sin\theta\,e^{ikz\cos\theta}\,\mathcal{J}_1(k\rho\sin\theta)\,\mathrm{d}\theta \\
=& \int_0^\pi \sin\theta\cos(kz\cos\theta)\,\mathcal{J}_1(k\rho\sin\theta)\,\mathrm{d}\theta + i\int_0^\pi \sin\theta\sin(kz\cos\theta)\,\mathcal{J}_1(k\rho\sin\theta)\,\mathrm{d}\theta
\end{aligned}
\tag{A23}
$$

$$
\begin{aligned}
\Gamma :=& i\int_0^\pi \sin\theta\cos\theta\,e^{ikz\cos\theta}\,\mathcal{J}_1(k\rho\sin\theta)\,\mathrm{d}\theta \\
=& -\int_0^\pi \sin\theta\cos\theta\sin(kz\cos\theta)\,\mathcal{J}_1(k\rho\sin\theta)\,\mathrm{d}\theta + i\int_0^\pi \sin\theta\cos\theta\cos(kz\cos\theta)\,\mathcal{J}_1(k\rho\sin\theta)\,\mathrm{d}\theta
\end{aligned}
\tag{A24}
$$





and

$$\Xi := \int_0^\pi \sin^2\theta \, e^{ikz\cos\theta} \, \mathcal{J}_0(k\rho\sin\theta) \, \mathrm{d}\theta$$


$$= \int_0^\pi \sin^2\theta \, \cos(kz\cos\theta) \, \mathcal{J}_0(k\rho\sin\theta) \, \mathrm{d}\theta + i\int_0^\pi \sin^2\theta \, \sin(kz\cos\theta) \, \mathcal{J}_0(k\rho\sin\theta) \, \mathrm{d}\theta. \tag{A25}$$

Introducing the transformation

$$\tilde\theta = \theta - \frac{\pi}{2} \quad \text{or equivalently} \quad \theta = \tilde\theta + \frac{\pi}{2}$$

such that

$$\sin\theta = \sin\left(\tilde\theta + \frac{\pi}{2}\right) = \cos\tilde\theta$$

and

$$\cos\theta = \cos\left(\tilde\theta + \frac{\pi}{2}\right) = -\sin\tilde\theta,$$

shows that the integrands of the imaginary parts are symmetric functions with respect to the value $\tilde\theta = 0$ in the interval $\tilde\theta \in \left[-\frac{\pi}{2}, \frac{\pi}{2}\right]$, so that

$$\mathrm{Im}\,\Phi = \int_{-\pi/2}^{\pi/2} \cos\tilde\theta \, \sin(-kz\sin\tilde\theta) \, \mathcal{J}_1(k\rho\cos\tilde\theta) \, \mathrm{d}\tilde\theta = 0 \tag{A26}$$

$$\mathrm{Im}\,\Gamma = \int_{-\pi/2}^{\pi/2} \cos\tilde\theta(-\sin\tilde\theta) \, \cos(kz(-\sin\tilde\theta)) \, \mathcal{J}_1(k\rho\cos\tilde\theta) \, \mathrm{d}\tilde\theta = 0 \tag{A27}$$

$$\mathrm{Im}\,\Xi = \int_{-\pi/2}^{\pi/2} \cos^2\tilde\theta \, \sin(kz(-\sin\tilde\theta)) \, \mathcal{J}_0(k\rho\cos\tilde\theta) \, \mathrm{d}\tilde\theta = 0, \tag{A28}$$

whereas the real parts do not vanish in general, as illustrated in Fig. (A1). Thus, introducing the abbreviations

$$f(\underline{x}, k) := \mathrm{Re}\left\{\int_0^\pi \sin\theta \, e^{ikz\cos\theta} \, \mathcal{J}_1(k\rho\sin\theta) \, \mathrm{d}\theta\right\} = \int_0^\pi \sin\theta \, \cos(kz\cos\theta) \, \mathcal{J}_1(k\rho\sin\theta) \, \mathrm{d}\theta \tag{A29}$$

$$g(\underline{x}, k) := \mathrm{Re}\left\{i\int_0^\pi \sin\theta \cos\theta \, e^{ikz\cos\theta} \, \mathcal{J}_1(k\rho\sin\theta) \, \mathrm{d}\theta\right\} = -\int_0^\pi \sin\theta \cos\theta \, \sin(kz\cos\theta) \, \mathcal{J}_1(k\rho\sin\theta) \, \mathrm{d}\theta \tag{A30}$$

and

$$h(\underline{x}, k) := -\mathrm{Re}\left\{\int_0^\pi \sin^2\theta \, e^{ikz\cos\theta} \, \mathcal{J}_0(k\rho\sin\theta) \, \mathrm{d}\theta\right\} = -\int_0^\pi \sin^2\theta \, \cos(kz\cos\theta) \, \mathcal{J}_0(k\rho\sin\theta) \, \mathrm{d}\theta, \tag{A31}$$





the measurable part (i.e., the real part) of the magnetic field can finally be written as

$$\tilde{\underline{B}}(k,\omega_0,\underline{x}) = \mathrm{Re}\left\{ B_0 \left[ i\alpha_0' f\,\underline{e}_\phi + \beta_0'\left(g\,\underline{e}_\rho + h\,\underline{e}_z\right)\right]\right\}. \tag{A32}$$

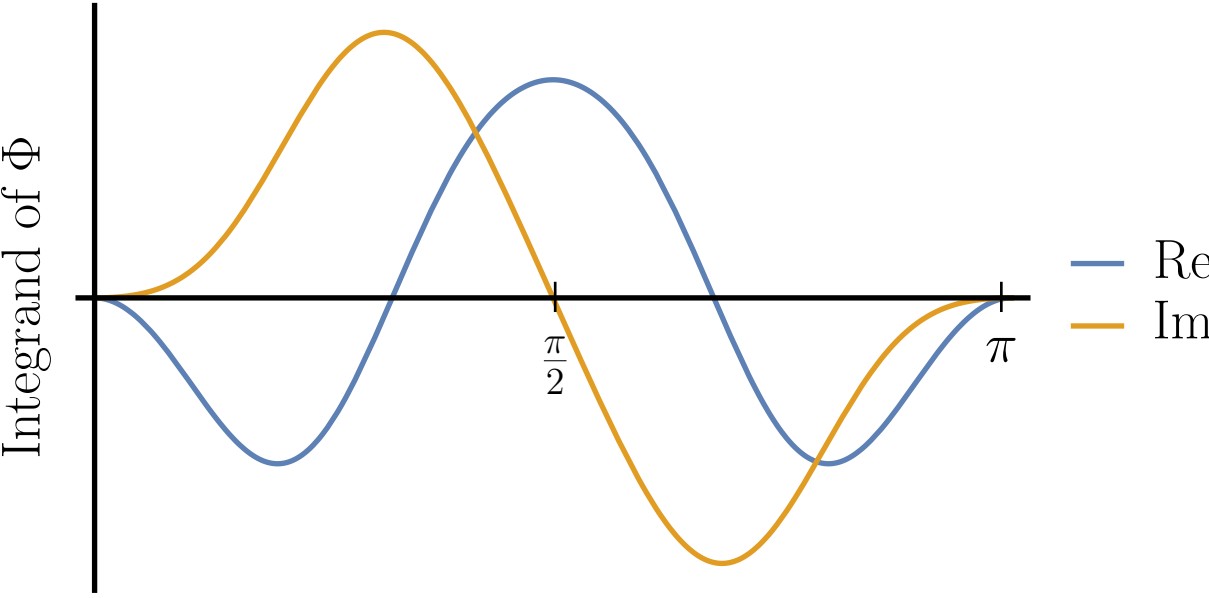

**Figure A1.** Real (blue) and imaginary part (orange) of the integrand of $\Phi$ in the interval $\theta \in [0,\pi]$.



*Author contributions.* All authors contributed equally to conception and design of the study; all authors read and approved the submitted version.

*Funding:* We acknowledge support by the German Research Foundation and the Open Access Publication Funds of the Technische Universität Braunschweig. The contribution by KHG was financially supported by the German Bundesministerium für Wirtschaft und Klimaschutz and the Deutsches Zentrum für Luft-und Raumfahrt under 50OC1803


*Competing interests.* The authors declare that they have no conflict of interest.

*Acknowledgements.* We are grateful to Norbert Fürstenau (DLR Braunschweig) to point our interest to electromagnetic knots.



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
