# Peer review of "Concerning the detection of electromagnetic knot structures in space plasmas using the wave telescope technique"

_Annales Geophysicae, 2023_

## Referee Comment (RC2)

**Referee Report for "Concerning the detection of electromagnetic knot structures in space plasmas using the wave telescope technique" by S. Toepfer, K.-H. Glassmeier and U. Motschmann**

In this paper the authors make a very interesting contribution to the methodology for analysing space plasma data by extending the wave telescope technique by combining it with a new set of exact solutions of Maxwell's equations. The wave telescope technique has been originally based on plane wave solutions of Maxwell's equations, but various authors have extended it to different solution spaces. In the present paper, the authors propose to use the localized solutions of Maxwell's equations presented in a paper by R. P. Cameron in 2018 as the basic solutions of Maxwell's equations to be used in the wave telescope method. This is a novel and innovative approach and the application to plasmoids presented in the paper seems to indicate that it has considerable future potential.

I find this paper to be scientifically of high quality and very well written. For example, I did not find any obvious typographical errors and I believe the mathematical derivations are also correct.

I have only one point of clarification that I would like the authors to address. The "classical" wave telescope methods are based on the superposition of solutions of Maxwell's equations that are mathematically complete, i.e. all functions within a certain function space can be represented as a sum or integral over the basis functions. In the discussion in the Introduction the authors seem to imply that Cameron's knot solutions also form a complete basis for the function space to be considered, but that is not obvious from either Cameron's original paper, nor does it become completely clear from the analysis presented in the paper. While it is clear that Cameron's solutions can be written as superpositions of plane wave solutions, it is not entirely clear to this referee that the inverse is also true, and it would have to be if one is really considering this to be a change of basis functions. This may, however, not be crucial for making the method applicable to data analysis, but a clarification might help readers who could be interested in using this method.

---

## Author Response (AR1)

**Reviewer 1:**

*This is an interesting study that extends the classical telescope technique for localized nonlinear structures, magnetic knots. Authors provides mathematical background for such an extension and discuss one possible application, plasmoids in the planetary magnetotails. I think the paper is well written as will be interesting to AnGeo readers. There is only one comment that I would like Authors to consider before publication. Any such analysis technique should have a certain threshold for magnetic field fluctuations that prevent the reconstruction of original magnetic field structure. Readers will be interested in some estimates of this fluctuation level (relative to the ambient field) for the plasmoid reconstruction.*

Reply: Thank you very much for reviewing the paper and for the positive feedback. We added this aspect accordingly within the manuscript (see p. 15, ll. 310–313).

**Reviewer 2:**

*In this paper the authors make a very interesting contribution to the methodology for analysing space plasma data by extending the wave telescope technique by combining it with a new set of exact solutions of Maxwell's equations. The wave telescope technique has been originally based on plane wave solutions of Maxwell's equations, but various authors have extended it to different solution spaces. In the present paper, the authors propose to use the localized solutions of Maxwell's equations presented in a paper by R. P. Cameron in 2018 as the basic solutions of Maxwell's equations to be used in the wave telescope method. This is a novel and innovative approach and the application to plasmoids presented in the paper seems to indicate that it has considerable future potential.*

*I find this paper to be scientifically of high quality and very well written. For example, I did not find any obvious typographical errors and I believe the mathematical derivations are also correct.*

*I have only one point of clarification that I would like the authors to address. The "classical" wave telescope methods are based on the superposition of solutions of Maxwell's equations that are mathematically complete, i.e. all functions within a certain function space can be represented as a sum or integral over the basis functions. In the discussion in the Introduction the authors seem to imply that Cameron's knot solutions also form a complete basis for the function space to be considered, but that is not obvious from either Cameron's original paper, nor does it become completely clear from the analysis presented in the paper. While it is clear that Cameron's solutions can be written as superpositions of plane wave solutions, it is not entirely clear to this referee that the inverse is also true, and it would have to be if one is really considering this to be a change of basis functions. This may, however, not be crucial for making the method applicable to data analysis, but a clarification might help readers who could be interested in using this method.*

Reply: Thank you very much for reviewing the paper and for the positive feedback. We discussed this aspect accordingly within the manuscript (see p. 9, ll. 181–188).

**Changes in the manuscript:**

- changes in the manuscript are marked in blue
- p. 9, ll. 181–188: We added a discussion about the differences between the knot structures and plane waves in the context of mathematical basis functions.
- p. 11, l. 227: We corrected the definition of the position vector.
- p. 15, ll. 310–313: We added a discussion about the influence of the magnetic field amplitudes for the plasmoid reconstruction.